# Predictors of self-management practices among diabetic patients attending hospitals in western Oromia, Ethiopia

**Dereje Chala Diriba**[1,2]◉*, **Tariku Tesfaye Bekuma**[3]◉, **Firew Tekle Bobo**[3,4]

**1** Department of Nursing, Wollega University, Nekemte, Oromia, Ethiopia, **2** School of Nursing, The Hong Kong Polytechnic University, Kowloon, Hong Kong, **3** Department of Public Health, Wollega University, Nekemte, Oromia, Ethiopia, **4** Australian Centre for Public and Population Health Research, Faculty of Health, University of Technology Sydney, Sydney, New South Wales, Australia

◉ These authors contributed equally to this work.
* dereje-chala.diriba@connect.polyu.hk

## Abstract

### Background

Diabetes Mellitus recognized as one of the emerging public health problems in developing countries. Self-monitoring needs to be individualized and should assist people with diabetes. This study aimed to assess the predictors of self-management practices among diabetic patients attending hospitals in western Oromia, Ethiopia.

### Methods

A facility-based cross-sectional study was conducted from November 2017 to February 2018 in hospitals located in western Oromia, Ethiopia. An interview was made with a total of 400 diabetic patients attending the diabetes center and admitted to ward in the study hospitals. The data were entered into Epi Info software version 3.5.4. Data analysis was made using a statistical package for the social sciences (SPSS) version 20. Odds ratio (OR) was used to show the association. The statistical significance was considered at P<0.05, and potential confounding variables were controlled using logistic regression. The analyzed data were presented in texts and tables.

### Results

From a total of 398 interviewed patients, 129 (32.4%) practiced diabetes self-management. About 63.6% of the study participants' self-management practice was good. Most 103 (79.84%) of those who practiced self-management were presented with one of diabetes mellitus-related complications. Logistic regression analysis results showed that merchants were about six times higher in self-management practice [AOR of 5.945 (1.177–30.027 at 95% CI)] and those having family support in diabetes practiced self-management 2.87 times than others [AOR of 2.835 (1.386–5.801 at 95% CI)].

**Data Availability Statement:** All relevant data are within the manuscript and its Supporting Information files.

**Funding:** Authors DCD TTB, and FTB received funding from Wollega University (https://www.wollegauniversity.edu.et/). Grant Number: WU: 109,161/Research1-26. The funder had no role in study design, data collection and analysis, decision to publish, or preparation of the manuscript.

**Competing interests:** The authors have declared that no competing interests exist.

**Abbreviations:** ADA, American Diabetes Association; AOR, Adjusted Odds Ratio; CDC, Centre for Disease Control; CI, Confidence Interval; DKA, Diabetes Ketoacidosis; DM, Diabetes Mellitus; IDF, International Diabetes Federation; OR, Odds Ratio; SBGM, Self-monitoring of Blood Glucose; SPSS, Statistical Package for the Social Sciences; WHO, World Health Organization.

## Conclusions

Compared to the findings of previous studies, diabetes self-management practices of the participants was good. The study participants regular physical activity, food intake, medication adherence, and foot self-examination were moderate. Two variables, being a merchant and having family support were found to be the predictors of self-management practices. Predictors of self-management should be considered to boost self-management practice.

## Introduction

Diabetes Mellitus (DM) refers to a group of common metabolic disorders that share the phenotype of hyperglycemia [1]. It is one of the four common non-communicable diseases causing major morbidities and mortalities [2–4]. World Health Organization (WHO) estimated that 422 million adults had diabetes in 2014 [5]. International Diabetes Federation (IDF) estimated this number to be 629 million by 2045. In Africa, the prevalence of diabetes adults between 20–79 years was 16 million in 2017 and projected to be 41 million in 2045. About 69.2% were undiagnosed. Africa attributes 77% of the deaths under 60 years to diabetes mellitus. The figure is the highest proportion in the world. International Diabetes Federation in 2015 estimated that 5.2% of Ethiopian adults had diabetes [6].

Self-management is the ability of the patient to deal with all that a chronic illness entails, including symptoms, treatment, physical and social consequences, and lifestyle changes [7, 8]. Since health care cost for the treatment of acute and chronic complications of diabetes is high [9], self-management is compulsory. Ethiopia is one of the low-income countries [10]. Thus, supporting patient self-management practice plays a key role in effective chronic illness care and improve patient outcomes. Effective management of diabetes requires predominantly self-directed practices, where the individuals become responsible for the day-to-day decisions related to controlling their disease [11–14]. The emerging evidence supports the implementation of practice strategies that are conducive to patient self-management and improved patient outcomes among chronically ill patients [15]. Implementation of self-management needs to be individualized, and people with diabetes should be assisted to understand the impact of medication, food and physical activity on blood glucose control. Adequate self-management can minimize the disease and disease-related complications, while the frequency of self-monitoring can be determined based on the individual's self-management goals [16]. The key goal of self-management is controlling blood glucose, improving quality of life and reduction of diabetes-complications [17]. A systematic review conducted in 2018 in Sub-Saharan Africa on self-management of Type 2 diabetes shows that patients rarely self-monitored their glucose levels; had low duration/frequency of physical activity; moderately adhered to recommended dietary and medication behavior and had a poor level of knowledge about diabetes complications [18].

Numerous studies conducted in different parts of Ethiopia shows slightly more than half of patients living with diabetes had good self-management practice [19–21]. Different cross sectional survey indicated that dietary and medication adherence of diabetic patients was low [20–22]. Internationally, only two-thirds of diabetic patients perform daily self-monitoring of blood glucose [17]. However, self-monitoring of blood glucose was good in Ethiopia [22]. Inadequate attention to diabetes center and lack of knowledge about self-management was reported as main factor in Ethiopia [23].

Self-management practice in clients with chronic diseases is essential to maintain good health while taking the medications. However, the practice may vary from client to client due to several factors. Different researches have investigated that age, current occupation, lack of awareness, absence of self-practice health education, years of suffering from DM, having family members suffering from the illness and lack of knowledge about the illness were the factors that affect the level of self-management behaviors [24, 25]. Besides, belief in treatment effectiveness, family support, self-efficacy, awareness about the disease and social support were also the factors that affect self-management practice [26, 27].

Despite the knowledge about the factors that affect self-management practice, there was no comprehensive study conducted on self-management of chronic illness especially DM covering the different hospitals located in western Oromia, Ethiopia, as far as the researchers' knowledge is concerned. This study thus focused on the assessment of self-management practice and its predictors in the study facilities. The findings of this study believed to give useful input for policymakers mostly in enforcing and establishing self-management practice interventions which, in turn, may encourage immediate health care providers to consider it in their routine care practices.

## Methods and materials

### Study design and setting

A hospital-based cross-sectional study was conducted among diabetic patients on follow-up at diabetic centers from November 2017 to February 2018 in public hospitals found in western Oromia, Ethiopia, namely Nekemte specialized hospital, Gimbi general hospital, and Nedjo general hospital.

### Source population

All diabetic patients on follow-up attending hospitals were considered as source populations.

### Study population

All diabetic patients attending diabetic centers and wards of the study hospitals during the study period were subjects of the study.

### Eligibility criteria

Known diabetic patients who visited the diabetic centers for follow-up and wards to receive care were included in the study while patients with diabetic emergencies like diabetic ketoacidosis and diabetic coma were excluded.

### Sample size determination

A single population proportion formula was used to determine the sample size. The proportion of patients who performed self-management practice (54.7% according to a study done at Nekemte referral hospital, Ethiopia in 2013) was considered in sample size calculation [19]. Marginal error between sample size and population parameter of 5%, and 95% confidence level, and 5% non-response rate was considered. A total of 400 patients with known diabetes mellitus participated in the study.

## Sampling techniques

All known diabetic patients visited the study hospitals for follow-up, and those admitted to the wards were taken into consideration. The average monthly client load was taken from the daily average DM client flow of the hospital and the registry book. Systematic random sampling was used. The interval was calculated at each hospital. The sample was allocated proportionally to the client flow of the respective facilities. The first client who arrived at the waiting area on the first day of data collection, and who met the eligibility criteria was taken as the first candidate for the study. This process continued until the desired sample size was attained.

## Data collection tools and methods

The data was collected directly by interviewing diabetic clients after getting informed consent. The questionnaire was prepared in English by modifying from different literature sources with similar areas of interest. It was translated from English to Afaan Oromo, a local language, and re-translated back to English to ensure consistency. The questionnaire was pre-tested in Dambidolo hospital. Three trained diploma nurses were used as data collectors under the close supervision of one B.Sc degree nurse, and the data were collected in a face-to-face interview. For the presence of co-morbid, we had observed the patent's folder. The only physician confirmed and recorded disease(s) was taken as a co-morbid disease.

## Data processing and analysis

Data were entered into Epi Info 3.5.4 software package and cleaned first. Then, the analysis was made using a statistical package for the social sciences (SPSS) software package version 20. Analysis of overall self-management practice was done by transforming the scores on closed-ended questions related to self-management practices. Using the odds ratio (OR) with a 95% limit of the confidence interval, the association of dependent and independent variables was analyzed, and their degree of associations was computed. Potential confounding variables were controlled using binary and multivariate logistic regression. Statistical significance was considered at $P<0.05$. Finally, the analyzed data were presented using frequency, percentage, and texts.

## Data quality control

A pre-test was conducted at Dambidolo hospital on 5% of the total sample size to check clarity, understandability, and consistency of the data collection tool. Then, the necessary amendments were made to the questionnaire before the full-scale data collection was implemented. Data collection was conducted under the close supervision of supervisors and the collected data were checked for completeness.

## Study variables

**Dependent variable.**  Diabetes self-management practice.

**Independent variables.**  Sociodemographic characteristics like sex, age, occupation, marital status, religion, level of education, lack of self-management education, patient education, getting family support, presence of DM-related complications and other health problems.

**Operational definitions.  Self-management**: The practice of diabetic patient's self-initiated and performed activities to control disease and maintain life, health, and wellbeing.

**Good self-management practice**:—Diabetic patients with average and above scores on closed-ended questions related to self-management practices.

**Poor self-management Practice**:—Diabetic patients with less than average score on closed-ended questions related to self-management practices.

**Hyperglycemia**—an abnormally increased concentration of glucose in the blood ($\geq$ 126 mg/dl at FPG).

## Ethics statement

Ethical approval was obtained from the Research and Ethics Committee of Wollega University. An official letter was written to each hospital to get official permission. Participants were informed that privacy and confidentiality were maintained. Written consent was taken from the study participants.

## Results

### Sociodemographic characteristics of the respondents

Table 1 depicts the overall characteristics of the study participants. A total of three hundred ninety-eight diabetic patients participated in this study, raising the response rate to 99.5%. More than half of the respondents were male (225, 56.5%). The average age of all respondents was 41.33 ± 18.93 (SD) years. Majority of the participants were married (255,64.1%), Oromo people (377, 94.7%), living in urban area (204, 51.3%) and Protestant Christians (219, 55.0%). The remaining (106, 26.8%) were single, 20 (5%) Amhara people, (194. 48.7%) living in rural areas and (136, 34.12%) followers of Orthodox Christian religion. One hundred eighteen (29.6%) respondents had attended college/university, whereas about 92 (23.1%) can not read and write, and only 20.4% had attended secondary school education.

Student patients were relatively high (21.6%) followed by housewives (20.4%) and government employees (20.4%). Based-on participants' believe, more than half, 54%, of the participants had a middle-level income compared to their neighbors. About 30.7% of the participants were poor and a very small proportion of the participants were very rich (0.3%). The average family size was 5 ± 2 (SD). About three fourths of the participants (74.6%) were husbands and wives in the family, followed by sons and daughters (22.9%). Most of the study participants (91.0%) reported that they need family support for disease treatment. About two-thirds of the participants (65.8%) received support from their families.

More than half of the participants, 220 (55.3%) did not know the type of diabetes they have. One hundred twenty-three (31.0%) of the participants had Type 1 diabetes, while the other had Type 2 diabetes. One hundred eleven (28.0%) had the disease for more than a decade, whereas about 72% live with DM for less than a decade. One-third of the study participants (66.8%) believe that they can cured of the disease, and the majority (71.6%) thought that the medication they were taking could cure them. Near half of the participants, 195 (49.0%) believe that health care providers had a good approach towards them. And two-thirds of the participants, 266 (66.8%) alleged that they would recover from the disease while others did not.

The majority of the participants (93.8%) had either of the following chronic illnesses: hypertension, cancer, edema, Human Immunodeficiency Virus/Acquired Immunodeficiency Syndrome (HIV/AIDS), dyspepsia and bronchial asthma. About three-fourth (74.1%) of them had diabetes-related manifestations and complications. Again, more than half of the participants had shock (55.9%) followed by diabetes ketoacidosis (52.9%). Some of the respondents also reported nerve (32.9%) and eye problems (23.4%) (Table 2).

Study participants had practiced different self-management interventions. They had also been taking medications and most of them had a monthly follow-up. About half (49.7%) of them had been taking insulin regimen followed by oral hypoglycemic agents (39.7%) for treatment and glycemic control. Only 42 (10.6%) were taking combined medications. The

**Table 1. Socio-demographic characteristics of the study respondents attending hospitals at western Oromia, Ethiopia, 2018 (n = 398).**

| Variables | Categories | Frequency (%) |
|---|---|---|
| Residence area (home town) | Urban | 204 (51.3) |
| | Rural | 194 (48.7) |
| Sex | Male | 225 (56.5) |
| | Female | 173 (43.5) |
| Religion | Orthodox | 136 (34.1) |
| | Protestant | 219 (55.1) |
| | Muslim | 39 (9.8) |
| | Catholic | 2 (0.5) |
| | Others * | 2 (0.5) |
| Ethnicity | Oromo | 377 (94.7) |
| | Amhara | 20 (5) |
| | Gurage | 1 (0.3) |
| Educational status | Can't read and write | 92 (23.1) |
| | Grade 1–4 | 41 (10.3) |
| | Grade 5–8 | 66 (16.6) |
| | Grade 9–12 | 81 (20.4) |
| | College/university | 118 ()29.6 |
| Marital status | Single (never married) | 106 (26.6) |
| | Married | 255 (64.1) |
| | Divorced | 5 (1.3) |
| | Widowed | 32 (8.0) |
| Occupation | Housewife | 81 (20.4) |
| | Government employee | 81 (20.4) |
| | Merchant | 50 (12.6) |
| | Student | 86 (21.6) |
| | Local drink seller | 8 (2.0) |
| | House servant | 1 (0.3) |
| | Daily laborer | 41 (10.3) |
| Patient to family relation | Husband/wife | 297 (74.6) |
| | Grandparents | 6 (1.5) |
| | Son/daughter | 91 (22.9) |
| | Sister/brother | 2 (0.5) |
| | Home servant | 2 (0.5) |
| Family economy versus neighbor | Very poor | 21 (5.3) |
| | Poor | 122 (30.7) |
| | Middle | 215 (54.0) |
| | Rich | 39 (9.8) |
| | Very rich | 1 (0.3) |
| Need for family support | Yes | 362 (91.0) |
| | No | 36 (9.0) |
| Ever supported by family | Yes | 262 (65.8) |
| | No | 36 (34.2) |

*Waaqefataa (it is a religion)

**Table 2. Diabetes-related complications among diabetic patients attending hospitals in western Oromia, Ethiopia, 2018 (n = 398).**

| Variables | Categories | Frequency (%) |
|---|---|---|
| Presence of DM-related complications | Yes | 295 (74.1) |
| | No | 103 (25.9) |
| Presence of shock at any time | Yes | 165 (55.9) |
| | No | 130 (44.1) |
| Diabetic ketoacidosis (DKA) | Yes | 156 (52.9) |
| | No | 139 (47.1) |
| Nerve diseases | Yes | 97 (32.9) |
| | No | 241 (67.1) |
| Eye problems | Yes | 69 (23.4) |
| | No | 226 (76.6) |
| Foot ulcer | Yes | 60 (20.3) |
| | No | 235 (79.7) |
| Kidney diseases | Yes | 54 (18.3) |
| | No | 241 (81.7) |
| Hyperglycemic hyperosmolar non-ketotic state (HHSS) | Yes | 19 (6.4) |
| | No | 276 (93.6) |
| Heart diseases | Yes | 8 (2.7) |
| | No | 287 (97.3) |

adherence to medication was varying. About the participants' diet intake, the participants had been consuming food three times a day with the different food menu. However, only one-third of them used the food menu. About half of the respondents used to consume vegetables followed by starch at lunchtime. At dinner, 39.2% of them used starch, while the others (43.0%) used vegetables. Close to three-fourths of the participants (72.4%) did not use to consume snacks, and starch was the most consumable food item. About half (52.0%) of them performed regular physical activity while the remaining did not (48.0%). Again, more than half of the respondents (57.8%) performed regular foot care, whereas 168 (42.2%) did not. Participants did not perform annual foot care check-up.

## Predictors of self-management practice

Overall, about 63.6% of the study participants self-management practice was good, while 36.4% self-management practice was poor. One hundred eighty-four (46.2%) of the participants knew about diabetes self-management management, of which 129 (32.4%) were practicing it. Cross-tabulation of the variables showed that those who are female, living in urban, married, rich and merchant tended to practice diabetes self-management. Females practice self-management more than males, 74.2% and 68% respectively. The magnitude of self-management practice was relatively higher among married clients (73.5%) than never married (66.7%). About 73.7% of participants from urban and 64.3% of rural had practiced self-management. No difference in self-management practice was observed between participants who had developed DM-related complications and who had not. Self-management practice was high in participants who believed DM is curable, than those who did not believe so. Participants who were merchants practiced more diabetes self-management, followed by daily laborers (79.2%). Those economically rich tended to practice more self-management than their poor counterparts (Table 3).

**Table 3. Cross-tabulation of selected variables with diabetes self-management practice amongst diabetic patients attending hospitals in western Oromia, Ethiopia, 2018 (n = 398).**

| Independent Variables | Self-Management practice | |
|---|---|---|
| | Yes | No |
| Sex | | |
| Male | 83 (68%) | 39 (32%) |
| Female | 46 (74.2%) | 16 (25.8%) |
| Residence | | |
| Urban | 84 (73.7%) | 30 (26.3%) |
| Rural | 45 (64.3%) | 25 (35.7%) |
| Marital status | | |
| Single | 40 (66.7%) | 20(33.3%) |
| Married | 86(73.5%) | 31(26.5%) |
| Others | 3(42.9%) | 4(57.1%) |
| Occupational status | | |
| House wife | 8(50%) | 8(50%) |
| Government employee | 39(73.6%) | 14(26.4%) |
| Merchant | 21(87.5%) | 3(12.5%) |
| Student | 31(63.3%) | 18(36.7%) |
| Daily laborer | 19(79.2%) | 5(20.8%) |
| Others | 11(61.1%) | 7(38.9%) |
| Relative economic status | | |
| Very poor | 11(78.60%) | 3(21.40%) |
| Poor | 15 (11.60%) | 17(53.10%) |
| Medium | 80(74.10%) | 28(25.90%) |
| Rich | 23(79.30%) | 6(20.70%) |
| Very rich | 0 | 1 (100%) |

The logistic regression analysis results indicated that only two variables, namely occupation and having family support had shown statistically significant association with the practice of self-management management in both binary and multiple logistic regressions. Accordingly, merchants were observed practicing self-management about six times higher than clients with other occupations with AOR of 5.945 (1.177–30.027 at 95% CI). Clients who had family support in DM-related care were again observed practicing self-management 2.87 times more than those who had no family support with AOR of 2.835 (1.386–5.801 at 95% CI). No difference was observed among other variables entered into the regression model (Table 4).

## Discussion

Among 398 study participants, 63.6% of them had good self-management practice. Self-blood glucose monitoring, physical activity, diet adherence, medication adherence, and foot care are the components of self-management practice. This finding is comparable with the finding from Northern Ethiopia [28]. The result of this study demonstrates that self-management practice of people living with diabetes mellitus attending hospitals in western Oromia is higher than the result of previous studies conducted in Nekemte referral hospital [19], Jimma University teaching hospital [21], and Harar, Ethiopia [20], and Lesotho [29]. This maybe due to day to day improvement of awareness of the disease management and cultural variation.

Physical activity believed as essential in controlling blood glucose. The result of this study revealed that more than half of participants practiced regular physical activity. The participants

**Table 4. Binary and multi-logistic regression of selected variables with diabetes self-management practice amongst diabetic patients attending hospitals in western Oromia, Ethiopia, 2018 (n = 398).**

| Variables | | Self-Management practice | | P-Value | COR at 95% CI | AOR at 95% CI |
|---|---|---|---|---|---|---|
| | | Yes | No | | | |
| Occupation | Gov't employee | 39(73.6%) | 14(26.4%) | .082 | 2.786 (0.878, 8.839) | 2.492 (0.745, 8.337) |
| | Merchant | 21(87.5%) | 3(12.5%) | .014 | 7.000 (1.476, 33.207) * | 5.945 (1.177, 30.027) * |
| | Student | 31(63.3%) | 18(36.7%) | .350 | 1.722 (0.551, 5.380) | 1.424 (0.384, 5.282) |
| | Daily laborer | 19(79.2%) | 5(20.8%) | .060 | 3.800 (0.947, 15.250) | 2.871 (0.658, 12.527) |
| | Others | 11(61.1%) | 7(38.9%) | .516 | 1.571 (0.402, 6.142) | 1.024 (0.244, 4.3) |
| | Constant | 8(50%) | 8(50%) | 1.000 | 1.000 | 1.00 |
| Ever family support | Yes | 97 (77%) | 29 (23%) | 0.003 | 2.718 (1.4, 5.28) * | 2.835 (1.386, 5.801) * |
| | No | 32 (44.8%) | 26 (55.2%) | 1.00 | | 1.00 |
| Presence of other health problems | Yes | 50 (73.5%) | 18 (26.5%) | .438 | 0.769 (0.395, 1.495) | 1.419 (0.612, 3.292) |
| | No | 79 (68.1%) | 37 (31.9%) | 1.00 | | 1.00 |
| Presence of DM related complications | Yes | 103 (70.1%) | 44 (29.9%) | .981 | 0.99 (0.45, 2.2) | 0.896 (0.38, 2.11) |
| | No | 26 (70.3%) | 11 (29.7%) | 1.00 | | |

*Statistically significant association

level of physical activity is greater than the finding of the study conducted in Harar, Ethiopia [20]. This discrepancy may be due to increased awareness regarding diabetic self-management and differences in the measurement tools. American Diabetes Association (ADA) recommends that physical activity performance five days/week that lasts for 30 minutes [30]. In this study, only half of the respondents self-reported as they perform regular physical activity which is inconsistent with the recommendation of ADA. Under performance of physical activity perhaps a lack of awareness about recommendations on physical activity and lack of infrastructures. ADA recommends diabetic patients should perform a comprehensive foot evaluation at least annually to identify risk factors for ulcers and amputations [31]. More than half of diabetic patients performed foot care. Foot care practice is less than the practice of community-dwelling Philippine diabetic patients [32], and adult patients attended Black lion hospital, Ethiopia [33]. On the other hand, foot examination and foot self-management were not consistent with recommendations of ADA [34]. This may be due to a variation in awareness creation, poor income, and increased age of the participants.

World Health Organization recommends the consumption of a healthy diet. A healthy diet includes consumption of fruit, vegetables, legumes, nuts, and whole grains; less than 10% of total energy intake from free sugars; 30% of total energy intake from fats and less than 5g of iodized salt [35]. American diabetes food pyramid recommends bread, rice, pasta, and rice as the first food menu, followed by vegetables and fruits [36]. The majority of the participants were not using the food menu which is inconsistent with the recommendations of WHO and ADA [35, 36]. The result of this study is unlike the results of the studies from Northwest Ethiopia [37], and Harar, Eastern Ethiopia [20]. Inconsistency on food menu may related with a level of awareness, the purpose of using food menu, absence of diabetes food education, and difference in data collection tools. This implies the participants had no awareness about food menu.

The vast majority of diabetic patients (74.1%) had diabetes mellitus-related manifestations and complications. More than half of the participants had diabetes-related shock and diabetic ketoacidosis. The prevalence of DM-related complications was higher than the results of the study conducted at Jimma University teaching hospital [21]. Nerve diseases were reported by

one-third of the study participants which was slightly higher than the study done in Jimma, Ethiopia. However, the reported DM-related eye problems were more prevalent than the result of the study conducted in Jimma university referral hospital [21]. High prevalence of eye problems may be be due to the public awareness about diabetes-related complication and lack of consistent health education. This implies that diabetes self-monitoring of blood glucose was poor in the participants of this study.

Being female, living in urban, married, rich, merchant and having family support were associated with self-management practice. Occupation and family support were tended to be the predictors of self-management. The finding of this study shares a similarity with that of studies done in Iran and Malaysia [26, 27] Family support was reported to increase adherence to self-management. Increase in self-management practice is comparable with the finding of the systematic review and meta-analysis which reported the social support significantly improved self-management [38]. However, these factors were inconsistent with factors reported by Amente, Belachew [19], Kaehaban, Hongsranagon [24] and Formosa and Muscat [39] from Ethiopia, Thailand, and Malta, respectively. This variation may be demonstrated due to day to day changes in diabetes education that increases awareness and culture variation.

The present study has some strengths. First, the response rate was high. Second, it tried to find out the predictors of self-management practice using appropriate data analysis methods. Limitations of the study were using unstandardized and validated tools.

## Conclusions

Compared to the findings of previous studies, the diabetes self-management practice of the participants of this study was good. The study participants' regular physical activity, food intake, medication adherence, and foot self-examination was moderate. Being a merchant and having family support were found to be the predictors of self-management practice. Randomized controlled trials involving the participants is needed to proof. Predictors of self-management should be considered to boost self-management practice.

## Supporting information

**S1 File. Study questionnaire English version.**
(PDF)

**S2 File. Study questionnaire Afaan Oromoo version.**
(PDF)

**S1 Dataset.**
(SAV)

## Acknowledgments

Our gratitude goes to the study facility administrators, service providers and participants for their collaboration and information. The authors would also like to thank colleagues who contributed their valuable suggestions throughout this research work.

## Author Contributions

**Formal analysis:** Tariku Tesfaye Bekuma.

**Investigation:** Dereje Chala Diriba.

**Methodology:** Firew Tekle Bobo.

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
