## [Decision Letter · Decision Letter 0]

31 Oct 2019

PONE-D-19-26947

Predictors of self-management practice among diabetic patients attending western Oromia hospitals, Ethiopia

PLOS ONE

Dear Dereje Chala Diriba

Thank you for submitting your manuscript to PLOS ONE. After careful consideration, we feel that it has merit but does not fully meet PLOS ONE’s publication criteria as it currently stands. Therefore, we invite you to submit a revised version of the manuscript that addresses the points raised during the review process.

The topic appears to be the most studied in most part of Ethiopia including Oromia region.

Abstract:

The methods are very short sighted to provide proper information. The result looks fragmented. The conclusion drawn not based on the finding?

Introduction:

This section included literature review from different part of the world including Ethiopia. I found this necessary to make a strong argument in the discussion section. However, the write up looks very inconsistent and juggling from the beginning to the end. You should follow introduction writing guide line. Since, the topic is highly researched in various part of Ethiopia I recommend authors to incorporate as many literature from Ethiopia as possible.

Methods:

I have noticed similar study had been conducted one year prior to this study in Nekemet, Western Oromia region. What is your rationale to Nekemete Hospital in this study?

How you operationally defined the study variables in this research? What is your dependent and independent variables?

How you developed the questionnaire? How you validate your questionnaire? Where you pretested the questionnaire?

Result:

The socio-demographic section seems to have included junk of information and redundancy. You must take account of socio-demographic variables than unrelated information. In general, the result section is full of in consist, in comprehensive and fragments of data. Most of the paragraph appears to have a copy and paste. I would recommend the authors to give appropriate time to evaluate the manuscript from the beginning to the end before submission to make sure the information built-in is clear and vivid information to understand.

Discussion:

As it has been said this topic is one of the most studied, hence the chance to get appropriate literature to make a strong argument on the finding is likely very high even in Ethiopia. However, the arguments are shallow, not based on evidence and weak. Why you cited literature which doesn’t have any similarity to the study setting, for instance America and others. This makes the discussion very shallow and inadequate.

What is the limitation of this study?

Conclusion:

How do you measure whether self-care is good or bad? There is no single statement in the methods section that clearly defined self-care and how it was measured. Overall, the conclusion not inferred from the analyzed result.

We would appreciate receiving your revised manuscript by Dec 06 2019 11:59PM. To enhance the reproducibility of your results, we recommend that if applicable you deposit your laboratory protocols in protocols.io, where a protocol can be assigned its own identifier (DOI) such that it can be cited independently in the future. For instructions see: http://journals.plos.org/plosone/s/submission-guidelines#loc-laboratory-protocols

We look forward to receiving your revised manuscript.

Kind regards,

Solomon Assefa Woreta

Academic Editor

PLOS ONE

**Journal Requirements:**

2. Thank you for including your ethics statement: Wollega University Institutional review Board

Please amend your current ethics statement to confirm that your named institutional review board or ethics committee specifically approved this study.

3. Our editorial staff has assessed your submission, and we have concerns about the grammar, usage, and overall readability of the manuscript.  We therefore request that you revise the text to fix the grammatical errors and improve the overall readability of the text before we send it for review. We suggest you have a fluent, preferably native, English-language speaker thoroughly copyedit your manuscript for language usage, spelling, and grammar.

If you do not know anyone who can do this, you may wish to consider employing a professional scientific editing service.  

Whilst you may use any professional scientific editing service of your choice, PLOS has partnered with both American Journal Experts (AJE) and Editage to provide discounted services to PLOS authors. Both organizations have experience helping authors meet PLOS guidelines and can provide language editing, translation, manuscript formatting, and figure formatting to ensure your manuscript meets our submission guidelines. To take advantage of our partnership with AJE, visit the AJE website (http://learn.aje.com/plos/) and enter referral code PLOS15 for a 15% discount off AJE services. To take advantage of our partnership with Editage, visit the Editage website (www.editage.com) and enter referral code PLOSEDIT for a 15% discount off Editage services. If the PLOS editorial team finds any language issues in text that either AJE or Editage has edited, the service provider will re-edit the text for free.

Please note that PLOS ONE does not copyedit accepted manuscripts and that one of our criteria for publication is that articles must be presented in an intelligible fashion and written in clear, correct, and unambiguous English (http://www.plosone.org/static/publication#language). If the language is not sufficiently improved, we may have no choice but to reject the manuscript without review.

4. Please include additional information regarding the survey or questionnaire used in the study and ensure that you have provided sufficient details that others could replicate the analyses. For instance, if you developed a questionnaire as part of this study and it is not under a copyright more restrictive than CC-BY, please include a copy, in both the original language and English, as Supporting Information.

5. Please ensure you have thoroughly discussed any potential limitations of this study, e.g. the self-reporting nature of data collection.

6. Thank you for stating the following financial disclosure: “No”   a) Please provide an amended Funding Statement that declares *all* the funding or sources of support received during this specific study (whether external or internal to your organization) as detailed online in our guide for authors at http://journals.plos.org/plosone/s/submit-now.     b)  Please state what role the funders took in the study.  If any authors received a salary from any of your funders, please state which authors and which funder. If the funders had no role, please state: "The funders had no role in study design, data collection and analysis, decision to publish, or preparation of the manuscript."

c) If the study was unfunded, please state "The author(s) received no specific funding for this work."

7. Thank you stating the following in your competing interests statement:

"No"

Please complete the competing interests section fully.  If NO authors have competing interests, please enter: "The authors have declared that no competing interests exist."

If Authors have competing interests please enter competing interest details beginning with this statement:

"I have read the journal's policy and the authors of this manuscript have the following competing interests: [insert competing interests here]"

8.  We note that you have stated that you will provide repository information for your data at acceptance. Should your manuscript be accepted for publication, we will hold it until you provide the relevant accession numbers or DOIs necessary to access your data. If you wish to make changes to your Data Availability statement, please describe these changes in your cover letter and we will update your Data Availability statement to reflect the information you provide.

**Additional Editor Comments (if provided):**

The topic appears to be the most studied in most part of Ethiopia including Oromia region.

Abstract:

The methods are very short sighted to provide proper information. The result looks fragmented. The conclusion drawn not based on the finding?

Introduction:

This section included literature review from different part of the world including Ethiopia. I found this necessary to make a strong argument in the discussion section. However, the write up looks very inconsistent and juggling from the beginning to the end. You should follow introduction writing guide line. Since, the topic is highly researched in various part of Ethiopia I recommend authors to incorporate as many literature from Ethiopia as possible.

Methods:

I have noticed similar study had been conducted one year prior to this study in Nekemet, Western Oromia region. What is your rationale to Nekemete Hospital in this study?

How you operationally defined the study variables in this research? What is your dependent and independent variables?

How you developed the questionnaire? How you validate your questionnaire? Where you pretested the questionnaire?

Result:

The socio-demographic section seems to have included junk of information and redundancy. You must take account of socio-demographic variables than unrelated information. In general, the result section is full of in consist, incomprehensive and fragments of data. Most of the paragraph appears to have a copy and paste. I would recommend the authors to give appropriate time to evaluate the manuscript from the beginning to the end before submission to make sure the information built-in is clear and vivid information to understand.

Discussion:

As it has been said this topic is one of the most studied, hence the chance to get appropriate literature to make a strong argument on the finding is likely very high even in Ethiopia. However, the arguments are shallow, not based on evidence and weak. Why you cited literature which doesn’t have any similarity to the study setting, for instance America and others. This makes the discussion very shallow and inadequate.

What is the limitation of this study?

Conclusion:

How do you measure whether self-care is good or bad? There is no single statement in the methods section that clearly defined self-care and how it was measured. Overall, the conclusion not inferred from the analyzed result.

Reviewers' comments:

Reviewer's Responses to Questions

**Comments to the Author**

1. Is the manuscript technically sound, and do the data support the conclusions?

Reviewer #1: Partly

Reviewer #2: Yes

Reviewer #3: Partly

2. Has the statistical analysis been performed appropriately and rigorously? 

Reviewer #1: Yes

Reviewer #2: No

Reviewer #3: Yes

3. Have the authors made all data underlying the findings in their manuscript fully available?

Reviewer #1: No

Reviewer #2: No

Reviewer #3: No

4. Is the manuscript presented in an intelligible fashion and written in standard English?

Reviewer #1: No

Reviewer #2: No

Reviewer #3: No

5. Review Comments to the Author

Reviewer #1: This is a study on self-management practice among diabetes patients in Ethiopian hospital setting. The authors do a great job in providing the big picture of diabetes in Ethiopian context and identifying those predictors of self-management practice. They followed a rigor procedure in their study design, analysis and reporting. This study adds a valuable information in the management of diabetes in western Oromia hospitals by providing some predictors for self-management practice.

This study has several issues that need to be addressed before being considered for publication.

• The introduction is well written but does not inform the reader about the importance of measuring the self-management practice among diabetes patient. Why we should seek to measure the self-management practice of diabetes among other chronic illness should be explained.

• What self-management is not described to a reader in the introductory section, rather the definition happened in a latter section of the manuscript (at the beginning of the discussion). This is not good without introducing the concept at the beginning but presenting the result and defining the term in the discussion.

• As the authors reviewed there are several studies conducted in Ethiopia (Harar, Nekemte, Addis Ababa). Therefore, why they are interested to do same topic in Ethiopia at western Oromia is not clear. The justification made as to why this study is conducted is not very convincing. For instance, 45% of participants at Nekemte referral hospitals had poor diabetes self-management though 54.3% of them had diabetic related knowledge, it worth doing a study why such level of practice with higher level of knowledge and what are the factors for such low self-care practice using other study design such as case-control than repeating same study in same place.

• The last section of introduction mentioned the absence of diabetic self-management education program in the country. However, the relevance of mentioning this concept in this section is not explained. The authors know this idea in advance and makes them even to forward a recommendation on establishing the education center. The claims are not placed properly in the context of the previous literature. Generally, the introduction should be re-written signifying the importance of doing this study in the light of “effective patient self-management is necessary to prevent adverse health outcomes of diabetes” among diabetes patients in lower-middle income countries including Ethiopia.

Reviewer #2: Abstract:

• Under method section number of study participants needs to be included

• Under result number of those who have either good or poor self care management needs to be indicated.

• The conclusion should be in line with the pertinent finding described under abstract

• Recommendation is not in line with conclusion.

Introduction

• Definition of self care management needs to be included.

• References needs to be in logical order.

Methods

The study setting is not well described.

Tools used to assess self care management was not clearly indicated.

Analysis method for self care management is not indicated.

Under ethical clearance future tense was used.

Result

• Standard deviation of mean age should be reported in plus or minus, not only plus

• Under table 1 you put asterisk on the other category of religion, but not define it below the table as footnote.

• Eye problems which is considered as one complication of DM is indicated by two different numbers.

• Rather than including other information better narration follows similar table.

• There is redundancy of narration about patients believe of diabetic treatment before and after table 2

• Under statements which is written above the title “predictors of self management” under result section, there is a statement which says figure 2, but there is no figure in the document.

• Better you include all variable you consider for logistic regression to table 3 with their respective confidence interval and odds ratio rather than duplicating it under table 4

• How you found at the end two predictor variable using logistic regression is not clearly explained.

• Revise your interpretation of logistic regression especially your comparison of occupation.

• How self care management is measured?????????? And classified as good and poor under the title “level of self care practice” are not clearly indicated.

Discussion

Your discussion needs revision. Implications given by the researcher is not satisfactory.

Conclusion

The conclusion is not in line with that of the title of the study and there are repeated copying and pasting of what is already described under result section.

Reference

Some of the references listed are not utilized for preparation of the manuscript

Finally competing interest and list of abbreviations/acronyms are not included to the manuscript

Reviewer #3: The authors tried to make allignment between title, objective, results and conclussion . However the manuscript need further revision and modification particularly the reporting format and language used during reporting.

6. PLOS authors have the option to publish the peer review history of their article (what does this mean?). If published, this will include your full peer review and any attached files.

Reviewer #1: No

Reviewer #2: No

Reviewer #3: Yes: Markos Desalegn Beyene

---

## [Author Response · Author response to Decision Letter 0]

24 Dec 2019

We really appreciate the genuine comments of the editors and reviewers. For the comments given by reviewers and editor, we reacted and corrected promptly. We are ready to correct if any further comments provided. The point by point response to reviewers commented was uploaded.

---

## [Editor Report · Decision Letter 1]

5 Feb 2020

PONE-D-19-26947R1

Predictors of self-management practices among diabetic patients attending hospitals in western Oromia, Ethiopia.

PLOS ONE

Dear Dr. Dereje Chala,

Thank you for submitting your manuscript to PLOS ONE. After careful consideration, we feel that it has merit but does not fully meet PLOS ONE’s publication criteria as it currently stands. Therefore, we invite you to submit a revised version of the manuscript that addresses the points raised during the review process.

I gain would like to applaud for putting together the effort to make the necessary improvement on the manuscript. Most of the comments and feedback that have been given by reviewers appears to be corrected and incorporated accordingly, but there are few issues need to be addressed in order to proceed to the next step. For instance, the definition used to describe knowledge has a preconceived notion. Did you evaluate knowledge based on individual question mean or the summation mean of the knowledge questions? Secondly, the argument used to discuss the finding seems to be so frail and not enough to provide vivid clarification. In general, this section needs to have literary supported evidence with clear reference or citation for your argument. For example, the first paragraph in the discussion section has compared and contrasted other studies with your current, but the statement used to ornate the argument read as “This could be related to change to diabetic education in different ways. Thus, sustainable scheduled diabetic education is very crucial to increase self-management practice.” Where did you get this information? Is this information research driven or based on the predetermined knowledge? Hence, all the argument you might reach need to have study based explanation or argument. Make sure you make the necessary changes in this section to eliminate the inconsistency.

We would appreciate receiving your revised manuscript by Mar 21 2020 11:59PM. To enhance the reproducibility of your results, we recommend that if applicable you deposit your laboratory protocols in protocols.io, where a protocol can be assigned its own identifier (DOI) such that it can be cited independently in the future. For instructions see: http://journals.plos.org/plosone/s/submission-guidelines#loc-laboratory-protocols

We look forward to receiving your revised manuscript.

Kind regards,

Solomon Assefa Woreta

Academic Editor

PLOS ONE

Additional Editor Comments (if provided):

I gain would like to applaud for putting together the effort to make the necessary improvement on the manuscript. Most of the comments and feedback that have been given by reviewers appears to be corrected and incorporated accordingly, but there are few issues need to be addressed in order to proceed to the next step. For instance, the definition used to describe knowledge has a preconceived notion. Did you evaluate knowledge based on individual question mean or the summation mean of the knowledge questions? Secondly, the argument used to discuss the finding seems to be so frail and not enough to provide vivid clarification. In general, this section needs to have literary supported evidence with clear reference or citation for your argument. For example, the first paragraph in the discussion section has compared and contrasted other studies with your current, but the statement used to ornate the argument read as “This could be related to change to diabetic education in different ways. Thus, sustainable scheduled diabetic education is very crucial to increase self-management practice.” Where did you get this information? Is this information research driven or based on the predetermined knowledge? Hence, all the argument you might reach need to have study based explanation or argument. Make sure you make the necessary changes in this section to eliminate the inconsistency.

---

## [Author Response · Author response to Decision Letter 1]

11 Mar 2020

Question 1: Did you evaluate knowledge based on individual question mean or the summation mean of the knowledge questions?

Response: Thank you for thorough evaluation and observation. We didn’t evaluate the diabetes knowledge score; however, we reported the general knowledge about diabetes management. Thus, we have removed from operational definition part.

Question 2. Secondly, the argument used to discuss the finding seems to be so frail and not enough to provide vivid clarification. In general, this section needs to have literary supported evidence with clear reference or citation for your argument.

Response: Thank you for your comment. We noted that our discussion is weak, thus we made modifications with implications of the results. We also cited the literature we have used.

---

## [Editor Report · Decision Letter 2]

17 Apr 2020

Predictors of self-management practices among diabetic patients attending hospitals in western Oromia, Ethiopia.

PONE-D-19-26947R2

Dear Dr. Diriba,

We are pleased to inform you that your manuscript has been judged scientifically suitable for publication and will be formally accepted for publication once it complies with all outstanding technical requirements.

With kind regards,

Wen-Jun Tu

Academic Editor

PLOS ONE
---

## [Editor Report · Acceptance letter]

22 Apr 2020

PONE-D-19-26947R2 

Predictors of self-management practices among diabetic patients attending hospitals in western Oromia, Ethiopia. 

Dear Dr. Diriba:

I am pleased to inform you that your manuscript has been deemed suitable for publication in PLOS ONE. Congratulations! Your manuscript is now with our production department. 

With kind regards,

on behalf of

Dr. Wen-Jun Tu 

Academic Editor

PLOS ONE